# Kinetics of Anti-SARS-CoV-2 Antibody Response Following Two Doses of the BNT162b2 mRNA Vaccine: A Japanese Single-Center Primary Care Clinic Report Involving Volunteers and Patients with Autoimmune Disease

Tomoko Sugiura [1,2,*], Hiroaki Sugiura [2], Hiroaki Kato [1], Yuko Nariai [1], Yuuki Mizumoto [1], Kozue Hanada [2], Rieko Takahashi [2], Yukari Hinotubo [2], Naoko Tanaka [2], Mutsumi Sasaki [2], Haruki Eguchi [3], Hiroki Kamino [1,4] and Takeshi Urano [1,4,5]

1   Department of Biochemistry, Faculty of Medicine, Shimane University, 89-1, Enya-cho, Izumo City 693-0021, Japan
2   Sugiura Clinic, 2-8-3, Kitahon-machi, Imachi-cho, Izumo City 693-0002, Japan
3   Eguchi Clinic, 6-43, Enya-Arihara-cho, Izumo City 693-0023, Japan
4   mAbProtein Co., Ltd. 89-1, Enya-cho, Izumo City 93-0021, Japan
5   Vaccines and Therapeutic Antibodies for Emerging Infectious Diseases, Shimane University, Izumo City 693-0021, Japan
*   Correspondence: sugiura-q@hi.enjoy.ne.jp; Tel.: +81-853-20-2127

**Abstract:** Despite the promising effectiveness of the coronavirus disease 2019 vaccination using an mRNA vaccine, the short efficacy duration and some poor responses to the vaccination remain major concerns. We aimed to clarify the monthly kinetics of the anti-SARS-CoV-2 spike receptor-binding domain antibody response after two doses of the BNT162b2 vaccine in a Japanese population. A chemiluminescent enzyme immunoassay (CLIA) and an enzyme-linked immunosorbent assay were used to measure the antibody levels in 81 Japanese adults (age, <65 years). The antibody levels increased 10-fold at 2–3 weeks following the second dose of BNT162b2 and declined thereafter to approximately 50%, 20%, and 10% of the peak levels at 2, 3, and 6 months, respectively. To compare the antibody titers among different groups, older adults (age, >65 years; $n = 38$) and patients with systemic lupus erythematosus (SLE, $n = 14$) were also investigated. A decline in the mean relative antibody titers was observed in older men compared with younger men and in patients with SLE compared with individuals aged <65 years. Although the antibody levels increased drastically following two BNT162b2 doses, they then declined rapidly. Furthermore, poor responders to the vaccination were observed. Repeated vaccinations are required to maintain high antibody levels.

**Keywords:** BNT162b2; COVID-19 vaccination; Japanese people; older men; systemic lupus erythematosus

## 1. Introduction

mRNA vaccines have been implemented worldwide following the outbreak of the coronavirus disease (COVID-19) pandemic. A two-dose regimen of the BNT162b2 vaccine by Pfizer-BioNTech confers 95% protection against COVID-19 [1]. In Japan, public vaccination began in February 2021, and according to the Ministry of Health, Labour, and Welfare of Japan, 79% of the total population (100 million people) had received two vaccine doses by February 2022. Furthermore, 50% of the total population received a third dose in May 2022. In addition to group vaccinations in public spaces, primary-care physicians administered individual vaccinations in Japan.

One major issue following the mRNA vaccination for COVID-19 concerns the vaccine's ability to confer long-term immunity. Several reports have provided evidence that indicates the antibody titers increase dramatically following the administration of two doses of the

mRNA vaccine; however, these titers then decrease rapidly [2–5]. As the antibody titers decrease, so does the clinical effectiveness of the vaccination [6]. Thus, repeated booster vaccinations within a short interval are crucial to prevent disease and community spread.

Another issue with regard to the COVID-19 vaccination concerns interindividual differences in vaccine immunity. Even in a healthy population, the range of antibody titers after vaccination is widely distributed. Some factors, such as male sex or alcohol consumption habits, may influence interindividual differences [7]. In certain groups, such as older adults and those with autoimmune diseases or cancer, some individuals have reportedly shown an inadequate immune response to the COVID-19 vaccination and were poor responders [7–13].

Detection systems for anti-SARS-CoV-2 spike receptor-binding domain (RBD) antibodies are readily available in the Japanese healthcare system. The chemiluminescent enzyme immunoassay (CLIA) produced by Abbott Laboratories is one such system used worldwide. Nevertheless, data on the normal range of the antibody titers for the SARS-CoV-2 spike RBD after two doses of the COVID-19 vaccination in healthy Japanese individuals are limited. A high antibody titer has been shown to correlate with solid immunity against COVID-19 [6], although antibody production reflects only a part of the vaccine immunity.

This study aimed to assess the monthly mean antibody titer in Japanese adults aged <65 years after the second dose of BNT162b2. Through converting antibody titers into a mean titer percentage in relation to Japanese individuals aged <65 years, we compared the antibody titers between different groups, such as older adults or patients with an autoimmune disease, regardless of the time of the titer determination after the second dose.

## 2. Materials and Methods

### 2.1. Study Design

This single-center prospective study, conducted in Izumo City, Shimane Prefecture, Japan, involved individuals and patients who had received SARS-CoV2-mRNA vaccinations with Pfizer-BioNTech BNT162b2 (first and second doses) from April to October 2021 at medical clinics and public spaces. During this period, approximately 3000 vaccinations were administered to 1500 outpatients and healthy individuals residing in the vicinity of the Sugiura Clinic.

Blood samples for antibody testing were collected four times: on the day of the second vaccination (one dose), 2–3 weeks after the second dose (peak), 3 months after the second dose, and 6 months after the second dose of vaccination. Individuals who underwent at least two blood collections at a different time, other than the four previously mentioned time points, were included in the study. Information on all of the participants was collected, including medical information, smoking and alcohol consumption habits, and side-effects of vaccination, such as fever and total IgG serum levels. The study was approved by the Ethics Committee of MEDIKS (Sapporo, Japan) with approval no. MX-4150FN-151448 and the Shimane University Institutional Committee of Ethics (Shimane, Japan) with approval no. 4913. Written informed consent was obtained from all participants.

### 2.2. Participants

Inclusion criteria comprised volunteers aged 20–65 years who had received a BNT162b2 vaccination at the Sugiura Clinic from 14 September to 31 October 2021 and who provided written informed consent to participate in this study. Those with a history of COVID-19 or those who had received immunosuppressive therapies were excluded. In total, 81 healthy individuals aged <65 years (39 men, 42 women) were enrolled. The mean age $\pm$ standard deviation (SD) was 42.2 $\pm$ 11.9 years. We also included 38 individuals (age, >65 years; men, 50%) who attended the Sugiura Clinic with chronic diseases, such as hypertension but excluded those with autoimmune diseases. The mean age $\pm$ SD of the older adult group was 78.2 $\pm$ 6.0 years, and this group had not received immunosuppressive therapy. Furthermore, 14 patients with systemic lupus erythematosus (SLE) (women, 100%) who attended the Sugiura Clinic and received immunosuppressive treatments were also recruited. All of

the patients with SLE were in remission, and the mean SLE disease activity index (SLEDAI) was 1.14 ± 1.45. The mean age ± SD in this group was 42.2 ± 11.9 years, and all of the patients had received a median dose of 5.6 ± 3.0 (range, 1–12) mg/day of oral prednisone. Other standard immunosuppressive treatments included hydroxychloroquine in 11/14 (78.5%) patients and tacrolimus in 10/14 (71.4%) patients. All of the participants, including the volunteers, older adults, and patients with SLE, provided their written informed consent prior to participation in this study.

### 2.3. Measurement of Antibody Titers

The anti-SARS-CoV-2 spike RBD antibody titers were assessed using two detection systems. The CLIA, from Abbott Laboratories (Chicago, IL, USA), is used worldwide and was obtained under contract by Nihon Rinsho, Inc. (Kyoto, Japan) for commercial antibody testing. This system detects the anti-SARS-CoV-2 spike RBD IgG using an ARCHITECT i2000R analyzer (Abbott Laboratories). The cutoff and upper detection limits were 50 and 80,000 arbitrary units (AUs)/mL, respectively. The second detection system was a sandwich enzyme-linked immunosorbent assay (ELISA) system (E-S-001), newly developed by mAbProtein, Inc. (Izumo City, Shimane, Japan) to detect anti-SARS-CoV-2 spike RBD IgG type antibodies in the serum. The lower detection limit was 0.4 AU/mL, which was determined by adding two standard deviations to the mean optical density (OD) obtained from 50 assays of the zero standard. All the samples in the study were measured using the CLIA (Abbott) and ELISA (mAbProtein) systems, with the antibody titers showing a significant correlation (r = 0.841, $p < 0.001$, Pearson's test).

### 2.4. Kinetic Graph

A program using the R language, https://CRAN.R-project.org/package=AntibodyTiters (accessed on 30 September 2021), enabled us to visualize the raw antibody data at any sampling point in the form of kinetic graphs. At least two blood sampling points are required to generate a kinetic graph. The kinetic charts showed increases and decreases in antibody titers for individuals generally or for different categories of individuals, such as those aged <65 years or ≥65 years.

### 2.5. Antibody Data Comparison

Following data collection, the monthly mean titers following two doses of BNT162b2 in Japanese individuals aged <65 years were determined. The raw data of antibody titers of older individuals and patients with SLE were changed to percentages relative to the mean titer of the same sampling time as the individuals aged <65 years. At least two blood samples were obtained per person, and the mean percentage was calculated using two or more raw antibody titers. To compare healthy individuals and different categories of individuals, the mean percentages of the relevant groups were assessed. Multiple regression analysis was performed to determine the factors influencing high and low antibody level percentages.

### 2.6. Statistical Analysis

The data for the groups were compared using a Student's *t*-test after confirming the raw data were normally distributed, whereas the inter-group data were compared using either Pearson's or Spearman's tests. All tests were two-sided, and a *p*-value < 0.05 was considered statistically significant. Statistical analyses were performed using GraphPad Prism, Version 9.3.1 (GraphPad Software, San Diego, CA, USA) software.

## 3. Results

### 3.1. Kinetics of Antibody Response

Figure 1 shows the changes in the antibody titers over time in volunteers and individuals aged <65 years. The data are presented as the geometric mean at each time of the blood collection. Although the baseline data at pre-vaccination are unavailable, individuals

who had not received the COVID-19 vaccination or who had never been infected with COVID-19 showed no detectable antibodies to the SARS-CoV-2 RBD; this was confirmed through examining 50 pooled sera from before the COVID-19 pandemic (data not shown).

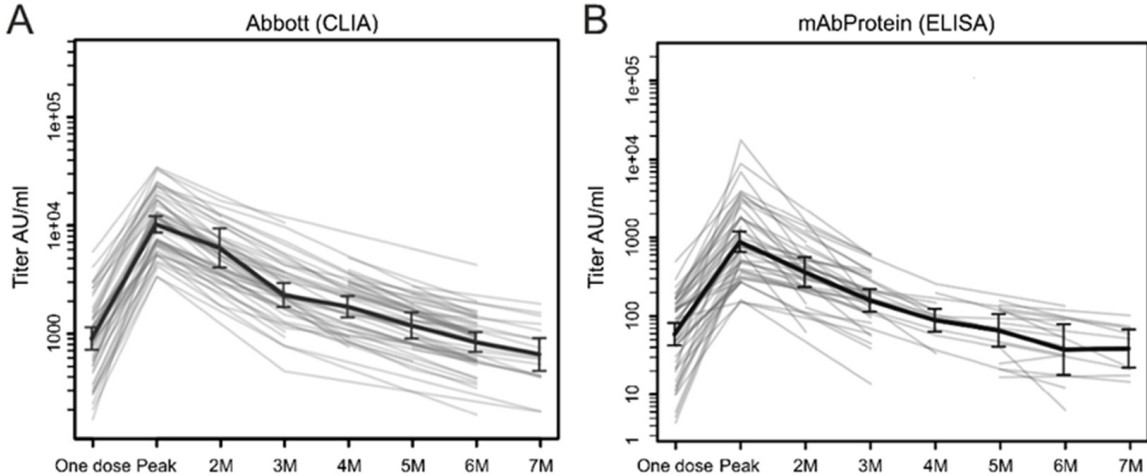

**Figure 1.** Kinetics of anti-SARS-CoV-2 spike RBD antibody response after two doses of the BNT162b2 vaccine. Antibody titers at each time of blood collection are presented as the geometric mean, as measured using (**A**) CLIA and (**B**) ELISA systems in individuals aged <65 years. *X*-axis, time of blood collection; *Y*-axis, anti-SARS-CoV-2 spike RBD S antibody titer (AU/mL); AU, arbitrary unit; CLIA, chemiluminescent enzyme immunoassay; ELISA, enzyme-linked immunosorbent assay; M, months.

The first blood samples were collected from individuals aged <65 years on the day of receiving the second dose (3 weeks after the first dose) and expressed as "one dose" (Figure 1). The mean ± SD and median (a 95% confidence interval [CI]) antibody titers were 1253 ± 1098 IU/mL and 917 (720–1202) IU/mL, respectively, using the CLIA system, and 96.4 ± 89.4 IU/mL and 77.2 (95% CI, 720–1202) IU/mL, respectively, using the ELISA system. The antibody titers showed a significant elevation 2–3 weeks after the second dose, with CLIA- and ELISA-derived means ± SD and median (95% CI) titers of 12,406 ± 8014 IU/mL and 10,983 (7171–13,045) IU/mL; and 1356.5 ± 1461.2 IU/mL and 759.1 (512.3–931.7) IU/mL, respectively. These peak titers were significantly higher (approximately 10 times) than those at time point 0 (a $p < 0.001$ with the Student's *t*-test for both methods of detection). However, the antibody titers measured using the CLIA system decreased rapidly to approximately 50% and 20% of the peak titers at 2 and 3 months, respectively, after the second dose. Subsequently, the antibody titers slowly dropped below 1/10 of the peak titers 6 months after the second dose. The monthly antibody titers measured using the CLIA and ELISA methods are presented in Table 1. As previously described, the blood samples for antibody testing were collected four times: one dose, peak, and at 3 and 6 months after the second dose. Those who underwent at least two blood collections at any time point were included in the study. Therefore, the monthly antibody titers were determined (Table 1).

*3.2. Interindividual Differences in Antibody Titers*

In this study, the Japanese individuals aged <65 years showed a considerable variation in their antibody titers. The peak antibody titers at 2–3 weeks after the second dose ranged from 3362 IU/mL to 34,386 IU/mL using the CLIA system, with an approximately 10-fold difference between the individual titers. Similarly, the peak antibody titers ranged from 150.1 IU/mL to 7160 IU/mL using the ELISA system, with an approximately 45-fold difference between the individual titers. Notably, individual differences in the antibody titers were maintained throughout the study period. Those with relatively high peak antibody titers tended to maintain higher titers at subsequent time points. A significant

correlation was noted between an individual's antibody titers at time 0 and the peak, at the peak and 3 months after vaccination, and at 3 and 6 months after vaccination (r = 0.768 by the Pearson's correlation test and a $p < 0.001$ for all the comparisons). Multiple regression analysis showed that a fever >37.5 °C, a major side-effect of the vaccination, occurred in 21 of 81 individuals (25.9%) and was related to higher antibody titers in individuals aged <65 years ($p = 0.03$) but was not otherwise related to age, smoking, alcohol consumption, sex, and serum IgG levels.

**Table 1.** Monthly mean and median titers of the anti-SARS-CoV-2 spike RBD antibody in individuals aged <65 years after receiving two doses of BNT162b2.

| Timing of Blood Sampling after the Second Dose | n | Abbott (CLIA) | | mAbProtein (ELISA) | |
|---|---|---|---|---|---|
| | | Mean ± SD (AU/mL) | Median (95% CI) (AU/mL) | Mean ± SD (AU/mL) | Median (95% CI) (AU/mL) |
| The same day (time point 0) | 50 | 1253 ± 1098 | 917 (720–1202) | 96.4 ± 89.4 | 77.2 (61.1–101.6) |
| 2–3 W after (peak point) | 53 | 12,406 ± 8014 | 10,983 (7171–13,045) | 1356.5 ± 1461.2 | 759.1 (512.3–931.7) |
| 2 M after (42–67 days) | 18 | 6030 ± 4332 | 5302 (2941–7337) | 363.8 ± 226.1 | 366.9 (201–524.3) |
| 3 M after (68–97 days) | 56 | 2322 ± 1751 | 2129 (1638–2409) | 163.9 ± 121 | 145.4 (117–172) |
| 4 M after (98–125 days) | 18 | 2305 ± 1204 | 2304 (1208–3061) | 98.2 ± 59.7 | 91.3 (60.4–153) |
| 5 M after (126–153 days) | 16 | 1296 ± 656 | 1083 (768–1734) | 83.7 ± 50.7 | 81.5 (59–133.3) |
| 6 M after (154–181 days) | 51 | 945 ± 662 | 888 (585–990) | 48.7 ± 34.6 | 42.7 (32.5–55) |

AU, arbitrary unit; CI, confidence interval; CLIA, chemiluminescent enzyme immunoassay; ELISA, enzyme-linked immunosorbent assay; M, months; SD, standard deviation; W, weeks.

### 3.3. Antibody Titers in the Older Population

Using the R program, we plotted the kinetics of the antibody response in the older adult population aged >65 years and compared them with that of healthy individuals aged <65 years (Figure 2). At the commencement of the project, the older adult population had completed two vaccination doses; therefore, there was no baseline data concerning their peak titers. The older adult individuals had relatively lower antibody titers than those aged <65 years nearly 3 months after the second vaccination.

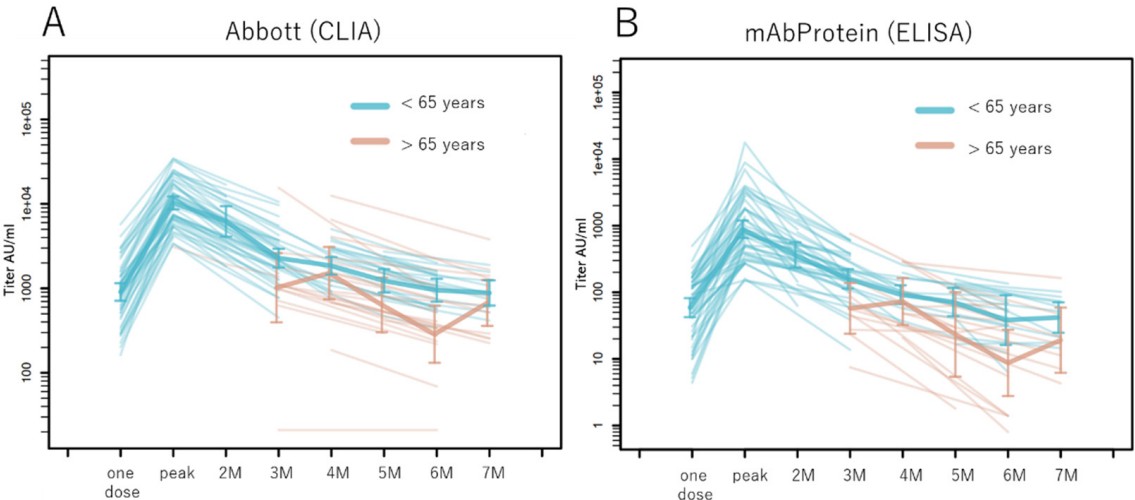

**Figure 2.** Kinetics of the anti-SARS-CoV-2 spike RBD antibody response in older adults. The geometric mean in individuals aged >65 (brown lines) and <65 (blue lines) years after two doses of the BNT162b2 vaccine measured using the (**A**) CLIA and (**B**) ELISA is shown in this figure. Data for individuals aged >65 years were obtained 3 months after they had received the second dose. AU, arbitrary unit; CLIA, chemiluminescent enzyme immunoassay; ELISA, enzyme-linked immunosorbent assay; M, months.

To compare the antibody titers of the older individuals with those of the young individuals, we changed all the raw antibody titers into percentages of the monthly mean titer (Table 1) of people aged <65 years, as previously described. The mean percentage of the antibody titers in the group aged >65 years (*n* = 38) was 84.6 ± 118.9%, which was comparable to that of the group aged <65 years (98.9 ± 67.4%) (Figure 3). However, when stratified according to sex, the mean titers were significantly lower in the men aged >65 years than in those aged <65 years (44 ± 59.8% vs. 94 ± 72.9%, difference: −49.9 ± 14.4%, 95% CI −78.5−−23.3, *p* = 0.0007, Student's *t*-test). Conversely, no significant differences were observed between the mean antibody titers in the women aged <65 years (104 ± 60.6%) and those aged >65 years (120 ± 145.3%) (Figure 3).

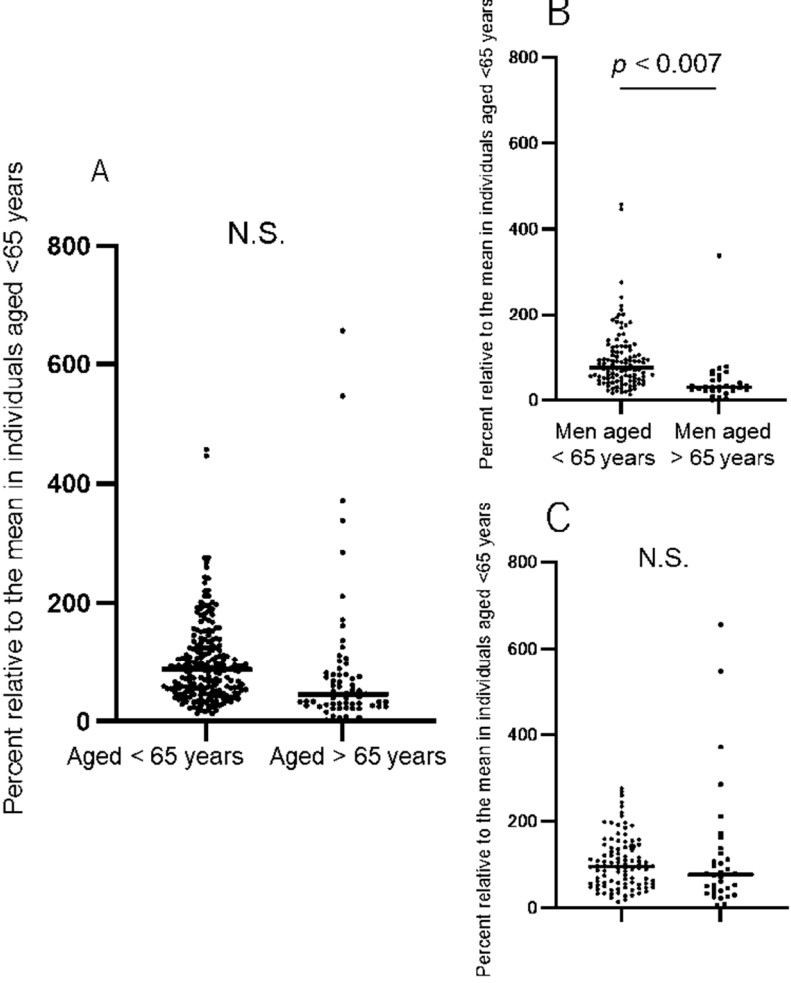

**Figure 3.** The anti-SARS-CoV-2 spike RBD antibody titers in younger vs. older individuals. Each antibody titer measured using the CLIA (Abbott) system was expressed as a percentage of the monthly mean antibody titer in individuals aged <65 years (as shown in Table 1). (**A**) The mean percent antibody titers in the individuals aged <65 years (98.9 ± 67.4%) and those aged >65 years (84.6 ± 118.9%) are comparable, with no significant difference. (**B**) When stratified according to sex, the antibody titers were significantly lower in men aged >65 years (44 ± 59.8%) compared with men aged <65 years (94 ± 72.9%) (Student's *t*-test, *p* = 0.0007; difference in means = −49.9 ± 14.4%; 95% CI −78.5−−23.3). (**C**) Contrastingly, there is no significant difference in the mean percent antibody titers between women aged <65 years (104 ± 60.6%) and women aged >65 years (120 ± 145.3%). N.S., not significant; CI, confidence interval; CLIA, chemiluminescent enzyme immunoassay; RBD, receptor-binding domain.

### 3.4. Immunosuppressed Patients

Fourteen patients with SLE who visited the Sugiura Clinic for immunosuppressive treatments were included in this study. All 14 patients were women aged <65 years (mean ± SD years, 42.2 ± 11.9) who had received a median dose of 5.6 ± 3.0 (range, 1–12) mg/day of oral prednisone. Other standard immunosuppressive treatments included hydroxychloroquine in 11/14 (78.5%) patients and tacrolimus in 10/14 (71.4%) patients. The blood collection times following the two vaccine doses were not the same for all patients; therefore, we expressed the antibody titers in the patients with SLE as a percentage of those in healthy individuals aged <65 years at the same time point. At least two blood samples were obtained per patient, and the mean of the percentages relative to individuals <65 years was calculated.

The mean percentage of anti-SARS-CoV-2 spike RBD antibody titers in patients with SLE was 58.2 ± 71.0%, which was significantly lower than that in individuals aged <65 years (98.8 ± 67%) (Student's *t*-test, $p = 0.0037$; the difference between the means, −40.7 ± 13.8%, 95% CI −68−−13) (Figure 4A). Notably, some patients with SLE showed only a slight increase in the antibody titers after the second dose, which rapidly declined below detection levels within 6 months. The antibody titers in 5/14 (35.7%) patients with SLE were <10% of those in individuals aged <65 throughout the study period.

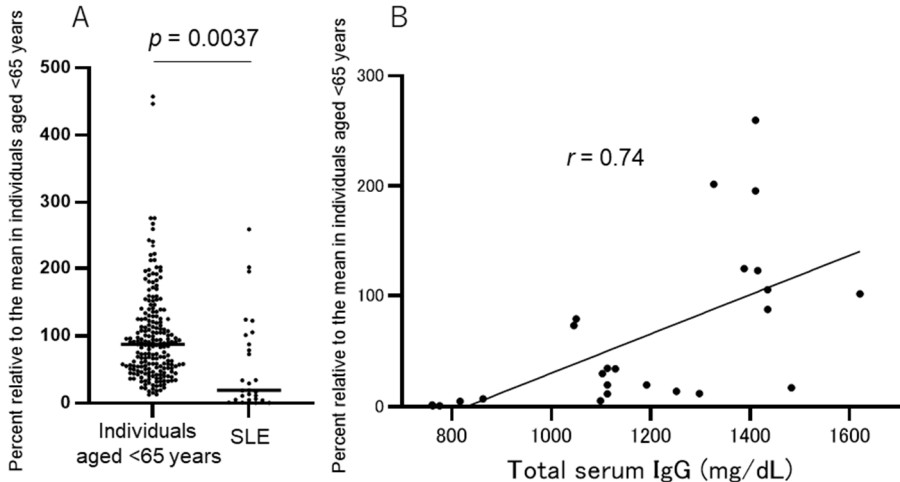

**Figure 4.** (**A**) The mean percent of antibody titers in patients with SLE (58.2 ± 71.0%) is significantly lower than that in individuals aged <65 years (98.8 ± 67%) (Student's t-test, *p* = 0.0037; difference mean titers: −40.7 ± 13.8%; 95% CI −68−−13). (**B**) A significant correlation between the percent antibody titers and total IgG levels in patients with SLE is seen (*r* = 0.75, 95% CI 0.51–0.88, *p* < 0.0001). CI, confidence interval; RBD, receptor-binding domain.

The correlation between the percentage of antibody titers and the dose of oral prednisone in these patients was of borderline significance (r = −0.577, *p* = 0.031, Spearman's correlation test). Furthermore, we assessed the total serum IgG levels (normal range, 800–1780 mg/mL) in patients with SLE, which are reflective of their immunosuppressive state. The mean total IgG in patients with SLE was 1168 ± 241 mg/dL, whereas that in individuals aged <65 years was 1241 ± 207 mg/dL; however, this difference was not statistically significant. In 2/14 patients with SLE, the serum IgG levels were lower than the normal limit, whereas they were within a normal range in healthy individuals. A strong correlation was found between the anti-SARS-CoV-2 spike RBD antibody titers and total IgG levels (r = 0.75, *p* < 0.0001, Spearman's correlation test; Figure 4B) in patients with SLE but not in individuals aged <65 years. We performed multiple regression tests to determine the factors associated with impaired antibody production after vaccination in patients with SLE. Only the serum total IgG levels showed an association (*t* = 2.76, *p* = 0.0157), whereas the prednisone dose, age, or other immunosuppressive agents showed no association.

## 4. Discussion

Our study design was similar to that of Naaber et al., who assessed the levels of anti-SARS-CoV-2 spike RBD antibodies in 122 Estonian healthcare workers using the CLIA system [4]. Consistent with their findings, we found a significant increase in antibody titers (10-fold in the present study vs. 19-fold in the study by Naaber et al.) after the second dose of BNT162b2, as well as a subsequent rapid decrease to 50% of the peak titer at 2 months and 20% at 3 months (in both studies). Subsequently, the antibody titers declined gradually to 1/10 (the present study) or 1/20 (Naaber et al.) of the peak titers 6 months after the second dose. The rapid decay of the antibody titers for up to 3 months and a gradual decline during the following 3 months have repeatedly been reported in other studies [2–6]. This pattern might reflect the initial humoral B and T-cell immune response generated through the two doses of the vaccine [14], the half-life of the human immunoglobulin [15], and the existence of memory B cells, which continuously produce small amounts of specific immunoglobulin for the SARS-CoV-2 spike RBD, as is the case following a COVID-19 infection [16,17].

Researchers have concentrated on investigating the duration of the neutralization activity shown through vaccination-induced antibodies. According to Naaber et al., the inhibition of the S protein of five variants of concern (wild-type, Alpha, Beta, Gamma, Delta, and Kappa) and the binding of the angiotensin-converting enzyme 2 (ACE2) receptor in an in vitro ELISA waned at 3 months after the second dose [4]. Another study involving 225 healthy Japanese medical workers who received two doses of BNT162b2 found that the half-life for neutralizing the activity to block SARS-CoV-2 virions from infecting target cells determined using immune sera was 67.8 days [3]. A test-negative study conducted from May to September 2021 evaluated the duration of protection from COVID-19 conferred through two doses of BNT162b2 in a large Israeli population [6]. During that study period, the Delta variant was prevalent in Israel, and an increased odds ratio of COVID-19 RT-PCR positivity was observed in individuals 146 days after BNT162b2 vaccination compared with that noted in individuals <146 days after BNT162b2 vaccination [6]. Based on these data concerning the Delta variant, a two-dose BNT162b2 vaccination appears to be effective for approximately 3–4 months.

The serum levels of the anti-SARS-CoV-2 spike RBD antibody and its neutralizing activity after vaccination reflect the different aspects of acquired humoral immunity; however, several studies have shown a close association between the two [3,4]. A high antibody titer equates to a high neutralizing activity. In our study, the mean antibody titer 3 months after the second dose was $2332 \pm 1751$ IU/mL with the CLIA and $163.9 \pm 121$ IU/mL with the ELISA. These values might indicate the minimum titer required for protection from COVID-19, particularly the Delta variant.

In this study, older adult men showed a poor response to the COVID-19 mRNA vaccination. Aging is a significant risk factor for severe COVID-19 [18]. Although several studies have reported an impaired immune response to the COVID-19 vaccine in older individuals [7,13], older COVID-19-recovered individuals show improved antibody and cellular responses, similar to younger individuals [13]. Therefore, booster doses are likely to be effective in the older population. Modifying the vaccination protocol, such as shortening the interval between doses or increasing the mRNA dose, might help improve the immune response in high-risk groups.

We selected SLE as a disease model. In a study involving 90 patients with SLE who had received two doses of BNT162b2, Izmirly et al. reported that 29% of the patients produced significantly lower levels of anti-SARS-CoV-2 spike RBD antibodies compared with the healthy controls [11]. Other investigators have reported that 13.2% of patients with autoimmune systemic diseases were non-responders to the COVID-19 vaccination [9]. In autoimmune inflammatory rheumatic diseases, immunosuppressive therapies, such as glucocorticoids, rituximab, mycophenolate mofetil, and abatacept, can impede the detection of anti-SARS-CoV-2-specific antibodies [10,19]. In the present study, 5/14 (35.7%) patients showed extremely low titers of anti-SARS-CoV-2 spike RBD antibodies, namely, <10% of

those in healthy individuals aged <65 years. Overall, the mean percentage of the antibody titers was <50% in patients with SLE than in healthy Japanese individuals aged <65 years. Initially, we focused on the association of the antibody titers with the prednisone dosage in these patients; however, a multiple regression model found that they were significantly associated with the total IgG levels. Low total IgG levels might be associated with impaired B cell function, primarily due to long-term oral prednisone [19], even at reduced doses.

The main limitation of our study was its small sample size. When we initiated the study, the two-dose vaccination regimen had already started for the Japanese population. Nonetheless, we obtained at least 50 samples for the time 0, the peak, 3 months, and 6 months after the second vaccination. Due to significant interindividual differences, even in the healthy younger group, the mean and median titers of the antibodies always showed a considerable difference.

**5. Conclusions**

Some concerns associated with COVID-19 vaccination using the mRNA vaccine BNT162b2 have been raised. This study found, first, that there was a rapid decline in the antibody titers after the second dose, requiring additional vaccinations at brief intervals. Second, some groups, such as older adult men or patients receiving immunosuppressive therapy, might be poor responders to vaccination. Finally, as booster vaccination has been reported to be effective in maintaining high antibody levels, modified protocols, such as applying shorter intervals or increasing the vaccine dosage for high-risk groups, might be required.

**Author Contributions:** Conceptualization, T.S. and T.U.; methodology, T.S.; software, H.K. (Hiroaki Kato) and Y.M.; validation, H.K. (Hiroki Kamino), H.S., and T.U.; formal analysis, T.S.; investigation, T.S.; resources, T.S., K.H., R.T., Y.H., N.T., M.S. and H.E.; data curation, H.S.; writing—original draft preparation, T.S.; writing—review and editing, T.U.; visualization, H.K. (Hiroki Kamino); supervision, H.S.; project administration, Y.N.; funding acquisition, T.U. All authors have read and agreed to the published version of the manuscript.

**Funding:** This work was supported by the Nippon Kayaku Co., Ltd. (to T.S.) and Shimane prefecture (to T.U.). The funders had no role in the design of the study; in the collection, analyses, or interpretation of data; in the writing of the manuscript; or in the decision to publish the results.

**Institutional Review Board Statement:** The study was conducted in accordance with the Declaration of Helsinki and approved by the Ethics Committee of MEDIKS (Sapporo, Japan) with approval no. MX-4150FN-151448. The date of approval was 15 September 2021.

**Informed Consent Statement:** Informed consent was obtained from all individuals involved in the study.

**Data Availability Statement:** The data presented in this study are available in the article.

**Conflicts of Interest:** T.U. is employed by Shimane University and is currently a co-founder and Chief Medical and Scientific Officer of mAbProtein, a biotech company focusing on the development and commercial utilization of mAbs for inflammation research, diagnosis, and treatment. All the other authors have declared no conflict of interest. The potential conflict of interest by T.U. does not alter the authors' adherence to all the journal's policies on sharing data and materials, as detailed online in the guide for authors.

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
