# Peer review of "Kinetics of Anti-SARS-CoV-2 Antibody Response Following Two Doses of the BNT162b2 mRNA Vaccine: A Japanese Single-Center Primary Care Clinic Report Involving Volunteers and Patients with Autoimmune Disease"

_2036-7449, doi:10.3390/idr15010003_

Round 1

Reviewer 1 Report

1. The title does not totally match with the content.  As the title saying "in the Japanese population" but actually is was done only in a small sample.

2. In LINE 28, you mentioned "men".  Does it mean male or you mean only "individuals"?

3. Unclear purpose about choosing the SLE group.

4. The inclusion and exclution criteria are unclear.

a. The group A is composed of young individuals (<65) without covid history.  Are they taking immunosuppressive therapy by chance? Are they having SLE or with family history?

b. The group B contains old individuals (>65) with no immunosuppressive therapy.  Can it be with SLE but without immunosuppressive therapy?

c. The group C includes only the patients with SLE.  Who were the participants diagnosing the SLE?  Are they at the same grading of serverity?  Is the type of immunosuppressive therapy affecting the results?

5. In LINE 90-91, "We enrolled 85 healthy individuals aged <65 years (39 men and 42 women)."  39+4285.

6. The explanations of figure 1, 3 and 4 are way too long to understand.  Also, the lines and dots in all figures are too similar to be differentiated.

7. Table 1 is good and clear to comprehend, but it is devided into 2 pages.

Reviewer 2 Report

The manuscript "Kinetics of anti-SARS-CoV-2 antibody response after two doses 2 of the BNT162b2 mRNA vaccine in the Japanese population" described long-term antibody response after vaccination with two doses of the Pfizer mRNA vaccine. The authors do an excellent analysis of antibody response up to 6 months after prime vaccination. The data presented are well organized and detailed. It would be of interest to perform a similar study evaluating a 3-dose regime for the mRNA vaccines to see if long-term antibody response improves. 

Author Response

Response: Thank you for your positive and encouraging review of our paper. 

Reviewer 3 Report

Line 88:  Add year

Line 103:  How were the detection limits defined?  Was this done via assay validation?  Are there similar upper and lower limits for the ELISA?  This needs to be stated either way.

Line 127:  Were data normal or normalized for the analyses?  Please confirm since the t-test is used for normal data distributions.

Line 137:  Please revise to "...vaccination and/or were not previously COVID-19 positive lacked detectable levels of..."  Similar language can be used at your discretion, but it currently reads as if the individuals "had COVID-19"

Line 138:  It is unclear what is meant by "via sampling examination."  Please elaborate.

Figure 1:  This figure is clear, but the figure caption needs attention.  Specifically, the abbreviations listed (beginning on Line 150) do not correlate with what is actually presented in the Figure.  For instance, it is understood that CI indicates "confidence interval", but CI is nowhere on the figure.  CI is in the figure caption, but can be defined in the text rather than in this fashion.  Please also define "AU"

Line 154:  Is this geometric mean?  If so, please state.

Line 154 and beyond:  The units on all these results is "IU/mL" whereas the figures specify AU/ml.  Consistency is needed.

Table 1:  This table requires a footnote/caption to define the various acronyms.  Also, please be consistent with the acronyms (e.g., "CI" vs "C.I".  Please also be consistent with the assay labels (Abbott vs. Abbott (CLIA) as in Figure 1).  Finally, it is unclear why the 2M, 4M, and 5M timepoints have such a smaller number of samples.  Please explain this in the text.

Line 173:  The age range of 20-64 years is not very narrow.  Please remove "narrow" or describe in another fashion.

Line 183:  How many individuals demonstrated fever?  Please include the number here.

Line 190:  The lack of baseline for the older population is indeed a weak point of the paper.  The data suggest that the older population had no response whatsoever.  Of interest (and perhaps this is not intentional) there appears to be a single individual in Figure 2B from the older population that appear to show a peak at 2M and a data point at "Peak".  Please confirm/clarify.  

Line 197 and throughout:  Please avoid the labels of "young and healthy" throughout the paper.  The use of "Under 65 years old" and "Over 65 years old" or similar is appropriate.  One cannot state that an individual is "healthy" based on an age range.

Line 199:  Same comment as Figure 1.

Figure 3.  Many of the points, particularly in B, are grainy/difficult to see.  Please clarify the figure.  Also, I do not think the Y-axis labels are correct (i.e., "Person relative to healthy young mean").  Please update the labels appropriately and avoid the use of labels including "healthy", "young", "elderly", etc.

Line 213:  Why were only the data from the CLIA system used?  Were the data trends the same via the ELISA?  This should be discussed.

Lines 301-305:  Although interesting, this paragraph is out of context and should be removed.

Line 306:  Please revise to "In our study..." or "In the present study..."

Lines 327-328:  This sentence should be a suggestive statement rather than a definitive statement.  The low IgG levels could have been due to a number of other factors as well.

Line 336:  The first sentence of your Conclusion is too broad.  Your study does not highlight problems associated with COVID-19 vaccination.  You were able to show waning immunity and age/sex differences following mRNA vaccination only.  Please perform a significant revision on this statement.

Line 220:  Same comment at Figure 1

Round 2

Reviewer 3 Report

No further comments.